# *MindReader*: Unsupervised Classification of Electroencephalographic Data

**DOI:** 10.3390/s23062971

**Published:** 2023-03-09

**Authors:** Salvador Daniel Rivas-Carrillo, Evgeny E. Akkuratov, Hector Valdez Ruvalcaba, Angel Vargas-Sanchez, Jan Komorowski, Daniel San-Juan, Manfred G. Grabherr

**Affiliations:** 1Department of Medical Biochemistry and Microbiology, Uppsala University, 75237 Uppsala, Sweden; 2Department of Cell and Molecular Biology, Uppsala University, 75237 Uppsala, Sweden; 3Science for Life Laboratory, Department of Applied Physics, Royal Institute of Technology, 11428 Stockholm, Sweden; akkurat@kth.se; 4Epilepsy Clinic, Instituto Nacional de Neurologia y Neurocirugía, Mexico City 14269, Mexico; dr.valdez.neurologia@gmail.com (H.V.R.); dsanjuan@innn.edu.mx (D.S.-J.); 5Independent Researcher, Guadalajara 44670, Mexico; ajvs_1687@hotmail.com; 6Washington National Primate Research Center, Seattle, WA 98121, USA; 7The Institute of Computer Science, Polish Academy of Sciences, 01-248 Warsaw, Poland

**Keywords:** electroencephalography, machine learning, precision medicine, unsupervised learning

## Abstract

Electroencephalogram (EEG) interpretation plays a critical role in the clinical assessment of neurological conditions, most notably epilepsy. However, EEG recordings are typically analyzed manually by highly specialized and heavily trained personnel. Moreover, the low rate of capturing abnormal events during the procedure makes interpretation time-consuming, resource-hungry, and overall an expensive process. Automatic detection offers the potential to improve the quality of patient care by shortening the time to diagnosis, managing big data and optimizing the allocation of human resources towards precision medicine. Here, we present *MindReader*, a novel unsupervised machine-learning method comprised of the interplay between an autoencoder network, a hidden Markov model (HMM), and a generative component: after dividing the signal into overlapping frames and performing a fast Fourier transform, *MindReader* trains an autoencoder neural network for dimensionality reduction and compact representation of different frequency patterns for each frame. Next, we processed the temporal patterns using a HMM, while a third and generative component hypothesized and characterized the different phases that were then fed back to the HMM. *MindReader* then automatically generates labels that the physician can interpret as pathological and non-pathological phases, thus effectively reducing the search space for trained personnel. We evaluated *MindReader*’s predictive performance on 686 recordings, encompassing more than 980 h from the publicly available Physionet database. Compared to manual annotations, *MindReader* identified 197 of 198 epileptic events (99.45%), and is, as such, a highly sensitive method, which is a prerequisite for clinical use.

## 1. Introduction

Biomedical signal measurement is a pivotal resource for assessment of patient well-being. As such, electroencephalogram (EEG) recording is a graphic portrayal of the difference in voltage between two different cerebral regions plotted over time [1]. EEG is a cornerstone in the assessment, treatment, and prognosis of neurological conditions. For example, epilepsy, a chronic disease of the brain that affects individuals of all ages worldwide, is characterized by epileptic seizures due to abnormal excessive or synchronous neuronal activity in the brain. This condition affects approximately 45.9 million patients worldwide with neurobiological, cognitive, psychological, and social consequences [2]. Additionally, EEGs are daily and widely recorded for diagnosis and follow-ups in critical and non-critical areas in hospitals and ambulatory facilities for multiple medical indications [1], producing large amounts of data.

EEG measures the electrical activity of the brain using electrodes uniformly placed on the scalp. This arrangement produces a multichannel recording of electrical fluctuations over time, where each channel is the product of the difference between potentials measured at two electrodes. In physiological terms, each channel captures the summed potential of millions of neurons. This allows EEG to make a physical two-dimensional representation of the brain electrical activity. As such, seizures, or epileptiform discharges, represent abnormalities of the brain’s electrical function.

The yield of the first scalp routine EEG recording to detect interictal epileptiform discharges (IEDs) after a first unprovoked seizure has low sensitivity in adults, ranging from 32% to 59% [3], and there is a small increase in the yield of IEDs if an EEG is performed within 24–48 h of a new-onset seizure [4]. Another strategy is to increase the diagnosis yield of EEG with serial EEG, long-term EEG, or sleep recordings [5], reaching a specificity of 78% to 98%, albeit at low sensitivity.

Traditionally, EEG interpretation is logistically challenging, especially 24/7, and requires highly specialized personnel, is time consuming, and additional training is needed. For these reasons, EEG is not feasible in many hospitals, while recording the EEGs themselves would be rather inexpensive. Automatically analyzing and annotating EEGs without the need for manual expert annotation thus would have great implications on the diagnosis, treatment, and outcome of patients in emergent or critical situations [6], even if it is just used as a filter to remove hours of uneventful recordings.

The wealth of information generated every day at health centers sparks the promise for automation of signal processing and interpretation by data-driven methods, either supervised, or semi-supervised. For example, machine-learning methods such as deep learning have been applied to detect arrythmias in electrocardiogram signals [7,8]. Likewise, studies report that deep learning algorithms achieve high accuracy on detecting drowsiness [9] and apnea [10] in EEG data.

Whether manual or automated, EEG seizure detection faces a number of challenges, including: (a) high biological variance among individuals; (b) presence of technical artifacts; (c) low incidence of IEDs in the recordings; (d) variability in the detection of abnormal events in different regions of the brain; (e) limited spatial resolution of scalp EEG recordings; and (f) long time sampling of the EEG recordings.

Currently, the reliability of visual analysis of EEG data is moderate [11,12,13]. Over-interpretation of normal waveforms as abnormal [14,15], inappropriate pattern-recognition of normal variants [16,17], and the use of subjective interpretation and reporting [11] constitute the main pitfalls [18].

In a recent review, Ahmad et al. 2022 compared a number of different methods for automated epileptic seizure detection, ranging from classic machine learning, e.g., support vector machines, recurrent neural networks, convolutional neural networks, and multi-layer autoencoder networks [19]. While some of these methods achieve high accuracy, they rely on a) the availability of large and accurately labeled training sets and b) computational power to perform training and classification. Moreover, most of these methods are “black boxes”, necessitating augmenting these algorithms with a component that explains the classifications [20].

*MindReader* devises an entirely different approach in that it (a) is unsupervised and therefore does not require any labeled training data; (b) is applicable to individual recordings from individual patients and is thus unbiased with regards to biological variation or differences in EEG recording procedures; (c) requires minimal computational power, making it ideal for analyzing many hours of recordings; and (d) while it does not address the issue of explainability, it focuses the attention of physicians on a small fraction of the recordings in which anomalies are observed. *MindReader* can thus largely automate the analysis process of EEG analysis, providing an alternative for the medical community to increase the yield of the diagnosis by utilizing an unsupervised machine-learning method. Thus, we aim at widening the availability of EEGs in neurological assessment, effectively shortening the time to receive the best health care for patients, facilitating health care providers’ workflow, and overall improving patients’ quality of life.

This study is organized as follow: we first detail and describe the dataset, method, and algorithm, followed by the evaluation on a large and annotated dataset. We then discuss our findings and devise future research to bring this methodology into the clinic to aid physicians and benefit patients.

## 2. Materials and Methods

### 2.1. Dataset Analyzed

We utilized a dataset from Physionet [21], a publicly available dataset of EEG recordings with annotations composed of 24 case recordings from 23 subjects (5 males, ages 3–22 years; and 17 females, ages 1.5–19 years, with average age 9.98 years, standard deviation 5.76). Case chb21 was obtained 1.5 years after case chb01, from the same female subject. The data per patient is divided among several files from 1 to 4 h long, recorded at 256 samples per second with 16-bit resolution, adding up to 686 files. The EEG montage is mostly bipolar (23 electrodes), with a few recordings containing 24 or 26 electrodes, as well as a unipolar montage.

### 2.2. Software

*MindReader* is an unsupervised machine-learning method that applies an autoencoder network for dimensionality reduction, a hidden Markov model for temporal segmentation, and a generative component for pattern characterization. *MindReader* is written in Julia, a modern, compiled, and efficient programming language for mathematical and scientific computing. Importantly, Julia offers convenient packages for preprocessing data (*FFTW*) and building artificial intelligence architectures, such as *Flux* [22], *POMDPs*, *DecisionTree*, and *NearestNeighborModels*. Furthermore, *MindReader* can be deployed via Docker, for reproducibility and portability, or directly by downloading the source code, available at https://github.com/DanielRivasMD/MindReader (accessed on 21 November 2022). The *MindReader* algorithm is displayed in Figure 1.

*MindReader* accepts European Data Format (EDF) files, a standard file format used for storage of medical time series [23]. *MindReader* is invoked through the script *ReadMind.jl*, where settings can be adjusted at the command-line interface. Alternatively, it can be integrated directly into scripting for processing of large datasets or even modified to admit other biomedical signals.

Workflow: *ReadMind* processes each recording individually and without any additional inputs, such as annotations. The output consists of a labeled segmentation for each channel so that similar patterns are assigned the same label. After loading the EDF files containing EEG recordings, each channel was binned into the overlapping windows of 256 samples that overlapped by 64 samples, followed by fast Fourier transform (FFT). We then applied an autoencoder neural network [24] that consisted of a single-layer encoder and a single-layer decoder, with the hidden layer containing 64 neurons (a quarter of the input/output), maintaining the same number of neurons on the decoder and the encoder [25]. We utilized the *leakyrelu* activation function, with a learning rate of 0.001 over 25 epochs. After training the neural network architecture on each dataset individually, we processed the input through the autoencoder and calculated the difference between the input and the output, i.e., the autoencoder error [26].

We then applied an algorithm based on a combination of a hidden Markov model and a generative component, typically a self-organizing map (SOM) to detect patterns in the EEG signal and generate hypotheses based on the data segmented by the HMM [27]. We used the publicly available library from https://github.com/DanielRivasMD/HiddenMarkovModelsReaders (accessed on 21 November 2022). We used the model with a penalty of 200, and a minimum frequency of 20.

### 2.3. Architecture

*MindReader* uses customized data structures to represent the different models and parameter settings. Importantly, each *MindReader* process is conducted per channel independently, which implies that:This is embarrassingly parallelizable for performance purposes;The computational complexity for a single channel is O(T), with T being the recording time, times O(N) with N denoting the number of channels. Even though memory consumption is low, it currently scales at O(T)*O(N), which can be further optimized, e.g., for deployment in embedded systems;*MindReader* is adaptable for different EEG montages, i.e., electrode placement;Identifying electrical anomalies independently allows for spatial localization per channel as well as hypothesizing the physiological relationship among different areas of the brain;Epileptogenic/irritative zones are potentially detectable and physically mappable. Notably, *MindReader* does not require specialized hardware and can be easily used after installation under any operating system: Linux, Windows, or OSX. Moreover, due to *MindReader*’s short run-time, it is potentially applicable in live interpretations.

## 3. Results

### 3.1. Physionet Dataset

The dataset we used for testing was composed of 24 case recordings, comprising more than 980 h recorded in 686 files, as detailed in Appendix A. The number of records per patient varied from 9 (minimum) to 42 (maximum), median = 29.5. A total of 198 seizures were recorded and manually annotated for 141 (20%) recordings. The distribution of seizures also varied across patients from 3 (minimum) to 40 (maximum), median being 6.5. The duration of seizures ranged from 6 to 752 s, with an average of 58.64 and standard deviation of 65.03, as shown in Figure 2. Seizures represented a small proportion of the recorded time, which is consistent with the literature. Overall, these events accounted for 0.33% of the total recorded time, i.e., 3 h, 13 min, and 31 s.

### 3.2. MindReader Predictive Performance

We evaluated the predictive performance of *MindReader* by comparing our model prediction to the annotated recording on each time frame, and calculated: (A) sensitivity; (B) specificity; (C) accuracy; (D) F1-score; (E) positive predictive value; (F) negative predictive value; (G) false positive rate; (H) false negative rate; (I) false discovery rate; (J) false omission rate; and K) Mathew correlation coefficient of our method, as detailed in Appendix A.

On a per-frame-of-recording basis, *MindReader* achieved an overall specificity of 81.52% at a sensitivity of 46.62%. Interestingly, case chb01 and chb21, which were recorded from the same subject at different time points, achieved similar specificity of 77.9% and 79.32%, respectively. If we consider that seizures only partially overlap with fixed-size recording frames and that sensitivity on a per-frame basis is thus a lower bound for practical purposes, we find that *MindReader* captured 197 of the 198 (99.45%) seizures recorded in the dataset. An example of *MindReader*’s output over the length of one recording is illustrated in Figure 3, for which the heatmap is represented with colors indicating states predicted by our model, ranging from low to high activity. The track of manual annotations is shown above the recording. On the top, the four time points of the raw recordings are shown, and a visual schematic of the EEG montage from the same time points is shown on the bottom.

Notably, subject chb12 had recordings with three different EEG montages, two bipolar and one unipolar. Importantly, the performance of *MindReader* was not affected in relation to the different montages, which suggests that our model can detect both bipolar and unipolar montages. Interestingly, subject chb11 presented an outlier in the duration of epileptic events in that it lasted 752 s, which was considerably longer than the rest of the seizures for the subject.

One feature of *MindReader* is that it processes signals from different electrodes independently, generating hypotheses for alternative states separately, so that events channels can be compared via the time stamps for in-depth analyses. We thus measured the number of simultaneous channels in which annotated seizures coincided, as illustrated in Figure 4. We observed that in most cases, annotated seizures were detected in more than one channel, with 99 (50%) seizures identified in all channels, 154 (78%) seizures identified in more than 80% of the channels, 181 (92%) seizures identified in more than 50% of the channels, 16 (8%) seizures identified in less than 50% of the channels, and only 1 (0.05%) not identified at all.

## 4. Discussion

While EEG interpretation is a time-consuming and highly-specialized task, it holds high clinical value for the diagnosis, treatment, and prognosis of neurological diseases. As such, EEG, as well as other biomedical signals, can benefit enormously from modern signal-processing techniques and unsupervised learning for automatic labeling.

Here, we present *MindReader*, a novel and unsupervised artificial intelligence method for anomaly detection applied to electrical epileptiform discharges on EEG signals. We tested the predictive performance of *MindReader* on the Physionet dataset, a publicly available and annotated dataset of EEG recordings, where *MindReader* achieved overall 81.52% specificity when measured per frame of recording, and detected 99.49% of manually annotated seizures, as described in detail in Appendix A.

We specifically designed *MindReader* to improve the efficiency of EEG analysis, and our findings indicate that the potential to automate signal detection in the clinic could dramatically reduce the time to patient attention and improve quality of life. Similarly, other methods were recently implemented to address clinical problems using machine- learning methods, such as cardiac arrhythmias [7]. Moreover, since the predictive performance of our algorithm is resilient to different montages, it can also be applied to other variants of EEG recordings, such as long-term recordings for patient monitoring and intracranial EEG. Moreover, our method likely generalizes to other, related medical signals, such as electrocardiography and electromyography, among others, given that those biomedical signals follow similar principles and are captured using similar techniques. More generally, we expect that our algorithm can form the basis to analyze and interpret any biological signal that is represented by variation of amplitude over time, where identification of anomalies in a timely manner is vital.

In contrast to supervised learning and machine-learning techniques, which are commonly used by deep neural networks, our method is unsupervised and learns from the data at hand, and as such, does not rely on previous annotations from experts. This is generally a big advantage, particularly in the case of EEG, where events are rare and reliable labeling is time-consuming, requires extensive training, and is expensive. Furthermore, since our method only relies on the individual biomedical recording itself and specifically on each channel independently, it will facilitate compensation for biological variance as well as methodological noise introduced during interpretation and minimize biases associated with labeling, e.g., over-interpretation.

Unlike other biomedical signals that are not based on measuring changes in electric voltage, where signals are periodic and harbor less variability, ECG, EEG, and EMG are highly variable as a function of the patient’s state of being and prone to biological inter-individual variance and methodological noise. Interestingly, our results show that the output from *MindReader* does not depend on the montage; thus, it could be used in combination with other methods [28]. Nevertheless, further testing with more case recordings is necessary to confirm these observations. Lastly, since *MindReader* is completely unsupervised, it could potentially also be used for other more invasive montages on EEG, such as subdural or stereo-encephalographic, where samples for training are sparse.

Our method identified the majority of seizures, effectively reducing by two orders of magnitude the recording space experts would need for verification. Further improvements can be achieved by evaluating other neural network architectures, for example variational autoencoders [29,30], denoising autoencoders [31], or other time-aware methods, such as bilateral long short-term memory neural networks [32,33], or attention-driven architectures, such as transformers [34,35]. By the same token, autoencoder neural network architectures with more layers have proven to be beneficial in terms of reducing the computational cost for function representation, even for small datasets, and yielding better compression [26]. Based on this unsupervised architecture, we will also further explore applying more elaborate post-processing filtering to refine the signal-to-noise ratio, and to better classify the detected alternative states.

## 5. Conclusions

*MindReader* devises a novel unsupervised machine-learning architecture capable of delivering an automatized complementary tool for speeding up EEG interpretation. Thereby, *MindReader* brings an innovative tool to aide many clinicians, not only highly specialized neurologists, for EEG and neurological assessment. Unlike deep learning, *MindReader* does not contain any hidden parameters: it takes a fresh look at each recording and learning from each new dataset. This, in turn, allows the expert to interpret the analysis results without having to speculate on what data the system had seen before, or was trained on.

Lastly, we stress that *MindReader*’s goal and purpose is to support clinicians rather than provide any diagnoses, thus complementing the physician’s expertise. In our view, we see this focus on supporting the clinician’s expertise while deriving information from only one recording at a time, one patient at a time, as the biggest strength of any unsupervised and generative method. While this comes with the inherent limitation of not being able to automatically provide ultimate diagnoses, we anticipate that such an approach is more likely to be adopted in the clinic over time.

## Figures and Tables

**Figure 1 sensors-23-02971-f001:**
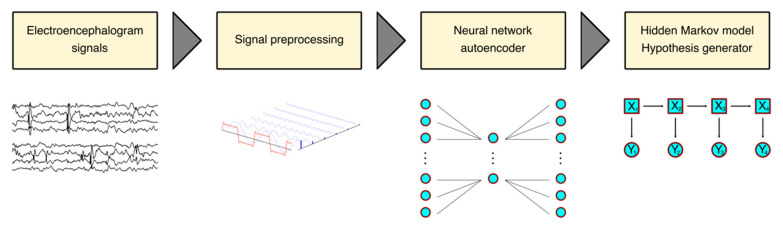
*MindReader* algorithm. From left to right, electroencephalographic (EEG) signals are loaded from standard European Data Format (EDF) files, then pre-processed by fast Fourier transform (FFT) and binned on overlapping windows. Next, signals are input to a neural network autoencoder where the autoencoder error is calculated, that is, the difference between the input and the autoencoder model post training is obtained, and considered anomaly. Finally, the signal is input to a hidden Markov model and hypothesis generator where states are assumed and labels are assigned. Importantly, the entire process is unsupervised and each channel is processed independently.

**Figure 2 sensors-23-02971-f002:**
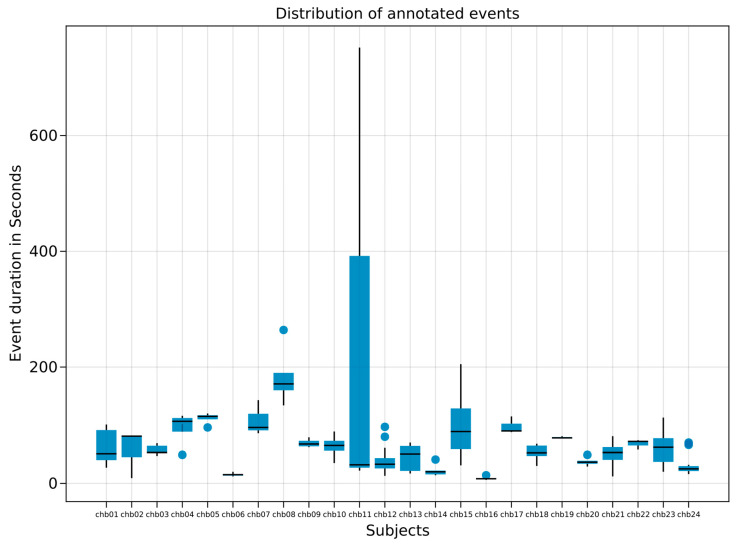
Boxplot of duration of annotated seizures per subject. Y axis indicates duration measured in seconds. X axis illustrates subject.

**Figure 3 sensors-23-02971-f003:**
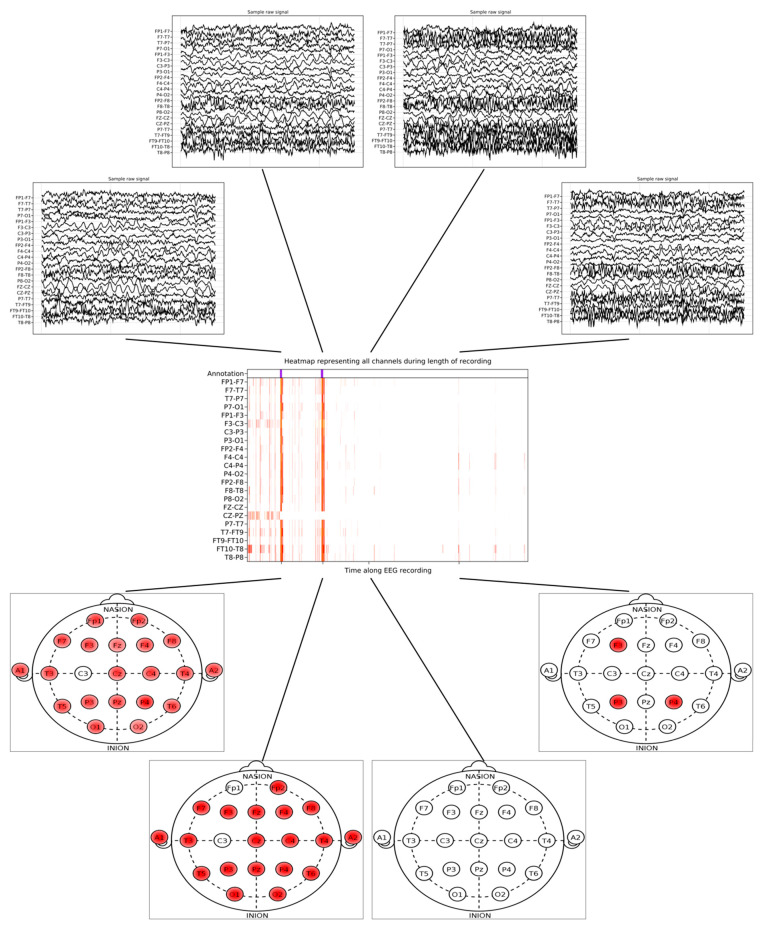
*MindReader* sample recording interpretation on subject chb04 (male 22 years old), record 28. Top plots illustrate *MindReader*’s output on EEG montage at four different time points during the recording. Middle heatmap shows *MindReader*’s hypothesized states along the recording per channel. Original Physionet manual annotation is indicated on top. Bottom plots display original EEG signals from same time points as interpretations.

**Figure 4 sensors-23-02971-f004:**
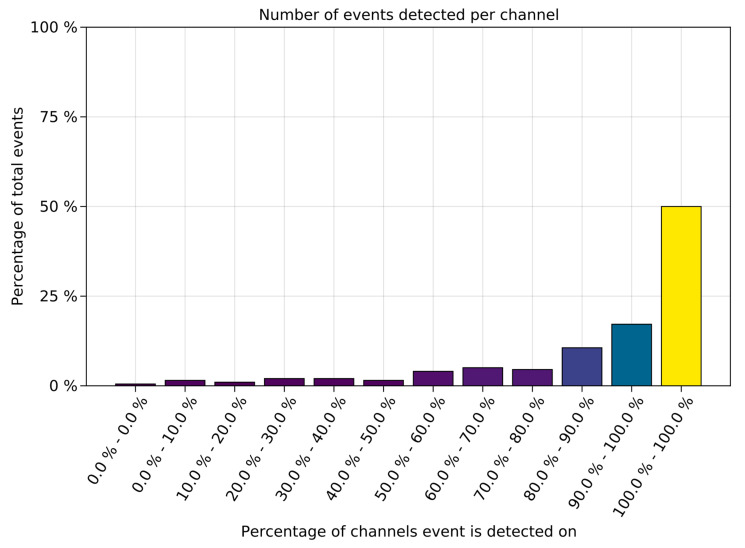
Barplot of seizure prediction as a percentage of total channels in the recording.

## Data Availability

The software is publicly available as source code at https://github.com/DanielRivasMD/MindReader (accessed on 21 November 2022) under the MIT License.

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
