# Peer review of "MindReader*: Unsupervised Classification of Electroencephalographic Data"

_sensors, 2023, doi:10.3390/s23062971_

Round 1

Reviewer 1 Report

The paper titled “Unsupervised Classification of Electroencephalographic Data” is developed to perform the detection of epilepsy. I have some major concerns to improve the readability and clarity of the manuscript.

1.      Your abstract does not highlight the specifics of your research or findings. Rewrite the Abstract section to be more meaningful.

2.      The phrase "various states of art models" is unclear, it is recommended to specify which models are being compared.

3.      Keywords should be in alphabetical order.

4.      Introduction section can be extended to add the issues in the context of the existing work and how the proposed algorithms/approach can be used to overcome this.

5.       In the Introduction section, the proposed method's new features and the results' main advantages over others should be clearly described.

6.      The problems of this work are not clearly stated. There is ambiguity in the statement understanding.

7.      More clarifications and highlights about the research gaps in the related works section needs to be included.

8.      If no one has proposed a method like the proposed algorithm before, this claim should be highlighted much more.

9.      A comparison with the state-of-the-art in the form of a table should be added along with remarks.

10.  Abbreviation should write in full-form of their first appearance in the text.

11.  Needs to verify script grammatically.

12.  Variables in equations are not explained properly.

13.  Full forms are repeated so many times.

14.  There are no significant curves observed in your script

15.  Discuss research contributions.

ü  Indicate practical advantages (in at least one separate paragraph),

ü  discuss research limitations (at least one separate paragraph), and

ü  supply 2-3 solid and insightful future research suggestions.

16.  Authors need to provide the merits of this study vs other review studies.

17.  Authors should add computational complexity.

18.  Limitations and the future scope should be added.

19.  Images quality is very poor.

20.  Conclusion should re-write with indication of potential results.

21.  Authors may remove unnecessary generalised well-known text from the script.

22.  Cite the following recent scripts

                                I.            https://doi.org/10.1016/B978-0-323-91197-9.00008-4

                             II.            https://doi.org/10.1155/2022/6486570

                          III.            10.1109/ICETET-SIP-2254415.2022.9791595

                          IV.            https://doi.org/10.1016/B978-0-323-91197-9.00010-2

Reviewer 2 Report

The authors presented a MindReader, an unsupervised method for signal anomaly detection applied to EEG signals. The method first processes the signal through an autoencoder neural network to detect EEG abnormalities. Then, temporal patterns are represented as a Hidden Markov Model, where different phases are identified and hypothesized as new states. The work has shown a MindReader’s predictive performance on 686 recordings, encompassing more than 980 hours from publicly available Physionet database. Results are interesting, as compared to manual annotations. The MindReader identified 197 of 198 epileptic events (99.45%). The paper has a good structure. However, major, and minor changes are needed, and as follows:

Major changes

1.      After lines #85, the authors did not explain the structure of the rest of the paper. For example, authors may write:  The rest of this paper is organized as follows. In Section …..

2.      The novelty of the work is unclear. Line #21 shows: “ we present MindReader, an unsupervised …”. Lines # 97-104 explain the proposed method shortly. I suggest adding more details about the method, such as details about the mathematical model, the signals being processed using Fast Fourier Transform, the neural network used. Also, lines #131-136 explains the used hidden Markov model. Thus, I suggest that the authors add more explanation about the method. This will help in addressing the novelty of the work.

Minor changes

1.      Please unify the font sizes in figures 2, 3, 4.

2.      Please add caption to the second table in the paper.

Reviewer 3 Report

The authors proposes autoencoder with HMM for automatic labelling of epileptic events. The result is interesting, however, the authors need to address the following issues:

1. The introduction section lacks the state-of-the-art in automatic seizure classfication.

2. The authors fail to convince why autoencoder is chosen as the neural network architecture in this paper although other architectures like LSTM, BLSTM or transformers are available in literature.

3. In a classification work like this one, supplementary Table 1 should be placed in the main article instaed of in the supplementary section.

4. As it is a binary classification, ROC and AUC should be calculated and reported for some represenatative subjects.

5. Table 1 should be placed as supplementary material.

6. Figure 3 need to be more legible.

7. For rigor, results need to be compared with other automatic seizure labelling techniques in the existing literature.

Round 2

Reviewer 1 Report

Author incorporated all my suggestions.

Reviewer 3 Report

Most of the suggestions are either addressed or answered.